

# The effects of knee extensor moment biofeedback on gait biomechanics and quadriceps contractile behavior

Amanda E. Munsch[1], Brian Pietrosimone[2] and Jason R. Franz[1]

[1] Joint Department of Biomedical Engineering, University of North Carolina at Chapel Hill and North Carolina State University, Chapel Hill, NC, United States of America

[2] Department of Exercise and Sport Science, University of North Carolina at Chapel Hill, Chapel Hill, NC, United States of America

## ABSTRACT

Individuals with knee joint pathologies exhibit quadriceps dysfunction that, during walking, manifests as smaller peak knee extensor moment (pKEM) and reduced knee flexion excursion. These changes persist despite muscle strengthening and may alter stance phase knee joint loading considered relevant to osteoarthritis risk. Novel rehabilitation strategies that more directly augment quadriceps mechanical output during functional movements are needed to reduce this risk. As an important first step, we tested the efficacy of real-time biofeedback during walking to prescribe changes of ±20% and ±40% of normal walking pKEM values in 11 uninjured young adults. We simultaneously recorded knee joint kinematics, ground reaction forces, and, via ultrasound, vastus lateralis (VL) fascicle length change behavior. Participants successfully responded to real-time biofeedback and averaged up to 55% larger and 51% smaller than normal pKEM values with concomitant and potentially favorable changes in knee flexion excursion. While the VL muscle-tendon unit (MTU) lengthened, VL fascicles accommodated weight acceptance during walking largely through isometric, or even slight concentric, rather than eccentric action as is commonly presumed. Targeted pKEM biofeedback may be a useful rehabilitative and/or scientific tool to elicit desirable changes in knee joint biomechanics considered relevant to the development of osteoarthritis.

## INTRODUCTION

Quadriceps function contributes to center of mass deceleration during the weight acceptance phase of walking (i.e., early stance) and facilitates homeostatic articular cartilage loading (*Lewek et al., 2002*; *Miyazaki et al., 2002*). Appropriate cartilage loading during gait is essential for maintaining health of mechanosensitive joint tissues, which may be negatively affected by excessive or insufficient repetitive loading (*Andriacchi et al., 2004*). However, individuals with knee joint pathology (e.g., unilateral arthroplasty, anterior cruciate ligament reconstruction [ACLR], or osteoarthritis) often exhibit persistent quadriceps muscle dysfunction that, at least in the case of ACLR, frequently persists long after return

Corresponding author
Jason R. Franz, jrfranz@email.unc.edu

to functional activity (*Benedetti et al., 2003*; *Fuchs et al., 2004*; *Roewer, Di Stasi & Snyder-Mackler, 2011*; *Noehren et al., 2013*). This dysfunction presents in the sagittal plane as smaller peak internal knee extensor moments (pKEM) and less knee flexion excursion during stance (*Lewek et al., 2002*; *Mizner & Snyder-Mackler, 2005*; *Roewer, Di Stasi & Snyder-Mackler, 2011*; *Sigward, Lin & Pratt, 2016*). Larger knee extensor moments have been found to correlate with more quadriceps force output and in turn greater compressive joint force (*Schmitz et al., 2017*). Accordingly, healthy individuals with typical pKEM values experience cartilage loading during walking that may protect against cartilage thinning—a factor considered relevant to osteoarthritis (OA) progression (*Schmitz et al., 2017*). In people with knee pathology, these aberrant patterns likely arise from some combination of quadriceps weakness (*Lewek et al., 2002*) and/or inhibition (*Blackburn et al., 2016*). However, while simple strength training can reverse asymmetric muscle weakness (*Devita, Hortobagyi & Barrier, 1998*; *Roewer, Di Stasi & Snyder-Mackler, 2011*), strengthening alone fails to alter more persistent and functional asymmetries in pKEM (*Devita, Hortobagyi & Barrier, 1998*; *Roewer, Di Stasi & Snyder-Mackler, 2011*; *Noehren et al., 2013*; *Sigward, Lin & Pratt, 2016* and/or knee flexion excursion (*Roewer, Di Stasi & Snyder-Mackler, 2011*; *Sigward, Lin & Pratt, 2016*). Novel strategies that more directly augment quadriceps output during functional movements are needed to restore physiological knee loading.

Biofeedback is a promising approach to cue changes in gait biomechanics that has been conducted in people with knee joint pathology. Most commonly, studies have used real-time biofeedback in people with ACLR and total knee arthroplasty to systematically alter vertical ground reaction forces (vGRF) during sit to stand and walking (*Zeni Jr et al., 2013*; *Luc-Harkey et al., 2018a*; *Luc-Harkey et al., 2018b*; *Christensen et al., 2019*). These studies have revealed insight relevant to the association between limb loading and, for example, biochemical markers indicative of cartilage mechanical responses. However, there is a growing need to use biofeedback to target root changes in quadriceps mechanical output during walking, which must overcome technical challenges associated with performing inverse dynamics calculations in real-time. Given that pKEM, a surrogate measure of quadriceps mechanical output during early stance, is reduced in individuals with knee joint pathology (*Devita, Hortobagyi & Barrier, 1998*; *Roewer, Di Stasi & Snyder-Mackler, 2011*; *Sigward, Lin & Pratt, 2016*), associates with less cartilage loading in contact force simulations (*Manal et al., 2015*), and persists following return to sport and despite strengthening (*Roewer, Di Stasi & Snyder-Mackler, 2011*), overcoming these challenges is important.

Quadriceps muscle forces are the largest contributor to knee loading during the early stance phase of walking (*Killen et al., 2018*). What we know about quadriceps muscle contractile behavior comes primarily from electromyographic measures and computational simulations. Those studies have in part reported on quadriceps activation amplitude, timing, and coactivation with other muscles spanning the knee during isolated contractions and functional movements (*Winter & Yack, 1987*; *Lass et al., 1991*; *Ivanenko, Poppele & Lacquaniti, 2004*; *Nyland, Klein & Caborn, 2010*; *Rice, McNair & Lewis, 2011*; *Arnold et al., 2013*). Based on their anatomical architecture and disproportionately high activation during weight acceptance (*Winter & Yack, 1987*; *Lass et al., 1991*; *Ivanenko, Poppele &*

*Lacquaniti, 2004*; *Arnold et al., 2013*), the quadriceps muscle–tendon units (MTUs) are most responsible for generating knee extensor moments in early stance. However, muscle activation alone need not associate with underlying MTU behavior (*Vigotsky et al., 2018*), and very few studies have empirically measured quadriceps muscle fascicle kinematics during functional activities such as walking. Accordingly, real-time biofeedback that targets pKEM in walking has significant added potential to improve our fundamental understanding of quadriceps MTU dynamics during weight acceptance and ultimately their role in knee loading.

Indirect evidence has perpetuated the textbook assumption that quadriceps muscles perform eccentrically during weight acceptance. Indeed, MTU lengthening is essentially prescribed by measured knee flexion excursion which, combined with relatively low compliance in proximal tendons, allude to active fascicle lengthening during early stance (*Ker, Alexander & Bennett, 1988*; *Farris & Sawicki, 2012a*; *Farris & Sawicki, 2012b*; *Manal et al., 2015*). However, the two studies to use dynamic ultrasound imaging to quantify quadriceps fascicle action *in vivo* during walking suggested that these muscles normally perform more isometrically during weight acceptance than previously appreciated (*Chleboun et al., 2007*; *Bohm et al., 2018*). Combining *in vivo* ultrasound with pKEM biofeedback—an approach designed to target quadriceps output—could accelerate our muscle-level understanding of quadriceps functional behavior and ultimately dysfunction in people with knee joint pathology.

As an important first step, our purpose was to apply real-time visual biofeedback of pKEM to uninjured walking participants to encourage changes in the quadriceps mechanical output while using ultrasonography to quantify vastus lateralis (VL) fascicle kinematics in the context of measured MTU length changes. We hypothesized that pKEM biofeedback would elicit prescribed increases and decreases in pKEM. We also hypothesized that the changes in pKEM would be accompanied by systematic changes in knee flexion excursion, VL MTU length change, and fascicle length change during weight acceptance, defined as the period between instants of heel-strike and pKEM.

## MATERIALS & METHODS

### Participants

Eleven uninjured young adults (6 females; mean ± s.d.; age: 23.6 ± 2.5 years, height: 1.7 ± 0.1 m, mass: 63.8 ± 9.3 kg) participated. Exclusion criteria included any history of knee joint surgery or major ligamentous injury, knee joint injury, or leg bone fractures in the previous six months, use of a lower extremity prosthesis, or other self-reported neurological or musculoskeletal condition that would limit walking ability. Methods and recruitment procedures for this study were approved by the Biomedical Sciences Institutional Review Board the University of North Carolina at Chapel Hill (18-2185). Each participant provided written consent prior to participation. Sample size was based on having 80% power to detect the smallest change in pKEM prescribed in this study (i.e., ±20%) compared to normative values from the literature (i.e., effect size = 0.77) (*Lewek et al., 2002*).

## Instrumentation

A 14-camera motion capture system (Motion Analysis Corporation, Santa Rose, CA, USA) sampling at 100 Hz recorded trajectories of retroreflective markers. Markers were secured to the anterior and posterior superior iliac spines, sacrum, lateral femoral condyles, lateral malleoli, posterior calcanei, and first and fifth metatarsal heads and an additional 14 tracking markers in clusters on the lateral thighs and shanks. A dual-belt, instrumented treadmill (Bertec, Columbus, OH, USA) recorded bilateral 3D ground reaction force (GRF) data at 1,000 Hz. We obtained participants' preferred overground walking speed using a photocell timing system (Bower Timing Systems, Draper, UT, USA). Photocells recorded the time taken for the participants to travel the middle three meters of a ten-meter walkway. Each participant's preferred speed was determined from the average of three overground trials (1.3 m/s $\pm$ 0.1) and used as the treadmill speed. Before walking trials commenced, participants acclimated to treadmill walking for five minutes. A 60 mm ultrasound transducer (LV7.5/60/128Z-2, UAB Telemed, Vilnius, Lithuania) recorded B-mode images through a longitudinal cross-section of participants' right VL. We placed the transducer midway between the greater trochanter and superior patella insertion (*Brennan et al., 2017*) and secured it with a custom flexible probe mount and elastic wrap. To confirm correct placement, we asked participants to flex and extend the knee while standing. We adjusted the probe location if this movement caused any out-of-plane motion. We collected cine B-mode images at 61 frames/s at a depth of 50 mm and used an analog signal indicating the start and stop of ultrasound image collection to synchronize with motion capture and GRF data.

## Experimental Protocol

This study used a real-time visual biofeedback paradigm to cue prescribed bilateral changes in pKEM during the weight acceptance phase of walking. Participants walked on the instrumented treadmill normally for two minutes. We immediately analyzed this trial using a real-time surrogate inverse dynamics model of the lower limb implemented in Matlab (Mathworks, Natick, MA, USA) to estimate baseline bilateral average pKEM values. Specifically, a custom Matlab script assumed a massless shank and foot and estimated the instantaneous right and left leg knee extensor moments from the cross product between the GRF vector and a position vector between the respective leg's lateral femoral condyle and the line of action of the GRF (Fig. 1A). pKEM values were extracted as the maximum value during the first half of stance. Using these baseline values, we established targets corresponding to −40%, −20%, +20% and +40% of normal pKEM values for use in subsequent biofeedback trials (Fig. 1B).

During trials with visual biofeedback, participants watched a video monitor positioned in front of the treadmill. The custom Matlab routine and inverse dynamics surrogate model previously used to derive target values estimated instantaneous bilateral pKEM for display in subsequent trials. The vertical position of a ball represented a moving average of instantaneous bilateral pKEM values over the previous four steps (Fig. 1B). The ordinate range for the display was set at ±60% of normal pKEM values for all participants. Before participants began to walk, we showed them a sagittal plane image of their retroreflective

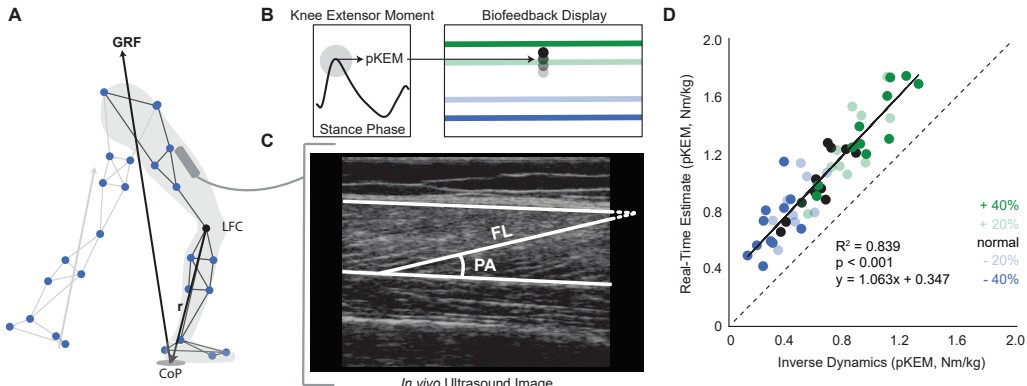

**Figure 1** **Real-time peak knee extensor moment (pKEM) biofeedback.** (A) We used a surrogate model to estimate peak knee extensor moment on a step-by-step basis as the cross product between the three-dimensional GRF vector and a position vector connecting the lateral femoral condyle (LFC) to the instantaneous center of pressure (CoP). (B) From these profiles, we used heel-strike events determined from the vGRF and extracted peak values from the first half of each stance phase to define pKEM. pKEM values were provided as biofeedback in the form of a moving average of the four most recent steps (i.e., two strides). While only one red horizontal target line was displayed as biofeedback, all four targets are included here and color coded by biofeedback trial for visualization. (C) We measured fascicle length and pennation at heel-strike and at the instant of pKEM. We calculated the pennation shown using two measurements: the angle between fascicle and image horizontal axis and the angle between deep aponeurosis and image horizontal axis. (D) Comparison of real-time estimates and post-hoc inverse dynamics estimates of pKEM. Dots represent an individual's average value across conditions indicated by color. Green and blue dots represent increases and decreases in pKEM compared to normal walking, respectively.

markers and GRF vector. We informed participants that changing the magnitude of the force between their feet and ground and/or changing knee flexion during early stance could affect the position of their pKEM values on the screen. We then started the treadmill and initiated the biofeedback paradigm, which displayed their instantaneous pKEM values from their previous four steps. All participants then completed a walking exploration trial without biofeedback targets in which they practiced varying their instantaneous pKEM values across the ordinate range (approximately one minute). During targeted biofeedback trials, the vertical position of a horizontal line on the screen indicated each target value (Fig. 1B). Specifically, participants completed one two-minute trial for each of four target values presented in random order. Finally, participants completed a static standing calibration and hip circumduction tasks (*Piazza & Cavanagh, 2001*) with additional markers placed on their medial femoral condyles and medial malleoli.

## Measurements and analysis

We filtered motion capture and force data using a low-pass Butterworth filter with a cutoff frequency of 12 Hz and estimated bilateral hip joint centers from static calibration and hip circumduction trials (*Piazza & Cavanagh, 2001*). We derived bilateral sagittal plane knee joint angles and VL MTU lengths via a global optimization inverse kinematics routine described in detail previously (*Hawkins & Hull, 1990*; *Silder, Heiderscheit & Thelen, 2008*; *Browne & Franz, 2019*). We estimated knee flexion excursion as the change in knee flexion angle between heel-strike and instant of pKEM. The routine then calculated bilateral
knee extensor moments using traditional inverse dynamics based on model kinematics, participant anthropometrics, and GRF data. We defined heel-strike with a 20 N vertical GRF threshold to obtain individual stride data and then assembled stride-averaged profiles from the second minute of each trial (~60 strides) for each outcome measure of interest. We report vGRF, knee flexion angle, and MTU data for the right limb to provide context for the fascicle data that was recorded unilaterally on the same limb.

We measured changes in VL fascicle length and pennation angle during weight acceptance from two strides acquired from the second minute of each trial. Here, we used UltraTrack, an open source ultrasound analysis routine in Matlab (*Farris & Lichtwark, 2016*). To ensure reliability, we opted to perform manual identification of fascicle lengths and pennation at specific keyframe events (i.e., heel-strike and the instant of pKEM) rather than automated tracking of kinematic time series, which can be susceptible to the accumulation of errors and require meticulous manual corrections. We used a 20 N threshold to identify the heel-strike frame in the vGRF data and found the local maximum in KEM stance data to identify pKEM frame. We manually identified an individual fascicle from deep to superficial aponeuroses at each of the two keyframe events for each stride. For fascicles that fell outside the image window, we defined the end of the fascicle based on its intersection with the linear projection of the aponeurosis (Fig. 1C), an estimation technique validated by *Ando et al., 2016* . In Ultratrack, the default pennation angle is measured with respect to the horizontal defined by the probe orientation. Accordingly, we manually identified the orientation of the deep aponeurosis neighboring the identified fascicle which we applied as a correction factor.

## Statistical analysis

Linear regression analysis evaluated correlation between real-time estimates and full inverse dynamic model of pKEM. Shapiro–Wilks tests confirmed all outcome measures were normally distributed. We include box and whisker plots showing outliers for all primary outcomes. We used a one-way repeated measures analysis of variance (ANOVA) with an alpha level of 0.05 to test for a significant main effect of biofeedback condition on six primary outcome variables: pKEM, knee flexion excursion, *peak* vGRF at the instant of pKEM, and *change* in VL MTU length, fascicle length, and pennation angle from heel-strike to the instant of pKEM. For outcome measures showing significant main effects of condition, we performed planned post-hoc pairwise comparisons to elucidate differences versus normal walking. One-sample t-tests also compared VL fascicle length change to 0 to characterize contractile state against isometric behavior. We report partial eta square ($\eta_{\text{p}}^2$) effect sizes from the ANOVA, and Cohen's d values for all pairwise comparisons.

## RESULTS

Participants produced $0.62 \pm 0.16$ Nm/kg pKEM when walking normally. Our real-time surrogate estimate of pKEM correlated well with that estimated via inverse dynamic calculations and, despite modestly overestimating those values, responded similarly to changes elicited using biofeedback ($R^2 = 0.839$, Fig. 1D). Indeed, targeted biofeedback elicited prescribed and predictable changes in pKEM (main effect, $p < 0.001$, $\eta_P^2 = 0.929$).

Pairwise comparisons revealed that participants produced 31% and 55% larger than normal pKEM when targeting 20% and 40% increases, and 25% and 51% smaller than normal pKEM when targeting 20% and 40% decreases, respectively ($p$-values $\leq 0.001$, $d \geq 1.066$, Figs. 2A, 2B). Participants walked normally with $16.8 \pm 3.5°$ of knee flexion excursion during weight acceptance and exhibited changes thereof in response to pKEM biofeedback (main effect, $p < 0.001$, $\eta_P^2 = 0.848$). For example, when cued to change pKEM by 40%, participants increased or decreased knee flexion excursion during weight acceptance by 30% and 36% respectively (pairwise $p \leq 0.001$, $d \geq 0.629$ Figs. 2C, 2D). pKEM biofeedback also elicited changes in vGRF (main effect, $p \leq 0.001$, $\eta_P^2 = 0.418$). Pairwise comparisons revealed that targeting a 40% change in pKEM elicited 9% greater or 5% less than normal peak vGRF (pairwise, $p \leq 0.037$, $d \geq 0.765$) (Figs. 2E, 2F).

During normal walking, the vastus laterals MTU lengthened by $1.21 \pm 0.26$ cm during weight acceptance—a change that differed significantly for all conditions (main effect: $p \leq 0.001$, $\eta_P^2 = 0.844$; pairwise: $p \leq 0.010$, $d \geq 0.428$). MTU lengthening increased by 20% and 34% when targeting 20% and 40% larger than normal pKEM, respectively. Conversely, MTU lengthening decreased by 10% and 17% when targeting 20% and 40% smaller than normal pKEM (Figs. 3A, 3B).

Despite VL MTU lengthening, VL fascicles shortened by $1.30 \pm 2.32$ cm during weight acceptance when walking normally. Changes elicited by biofeedback were modest and not significant(main effect: $p = 0.053$, $\eta_P^2 = 0.204$), and, unlike for MTU lengthening, no condition elicited behavior that differed significantly from isometric (one-sample $t$-test: $p \geq 0.092$, Fig. 3C, Table 1). During normal walking, VL fascicle pennation increased by $3.1 \pm 3.3°$ during weight acceptance. Similar to those in VL fascicle length, changes in VL fascicle pennation during weight acceptance were not significantly affected by pKEM biofeedback (main effect: $p = 0.056$, $\eta_P^2 = 0.202$, Fig. 3D).

## DISCUSSION

We aimed to test the efficacy of real-time visual biofeedback to modulate peak knee extensor moments—, herein used as a surrogate for quadriceps output—, during walking while quantifying associated changes in VL muscle fascicle kinematics in uninjured, young adults. Knee extensor moment profiles estimated using inverse dynamics calculations resembled those in the literature in timing and magnitude (*Besier et al., 2009*; *Noehren et al., 2013*). Moreover, our real-time surrogate model provided pKEM values consistent with those established from conventional inverse dynamic estimates. Consistent with our hypothesis, biofeedback elicited predictable changes in pKEM in uninjured young adults, augmenting step-to-step values during weight acceptance. These changes were accompanied by concomitant changes in knee flexion excursion. Furthermore, and consistent with joint kinematics, the VL MTU lengthened with the rise in pKEM during weight acceptance as hypothesized. However, contrary to our hypothesis, active VL muscle fascicles did not exhibit lengthening during early stance. Rather, our data suggest that the VL performs relatively isometrically, or even slightly concentrically, to accommodate weight acceptance in walking, not eccentrically as is commonly assumed. Together, our results: (1) allude to

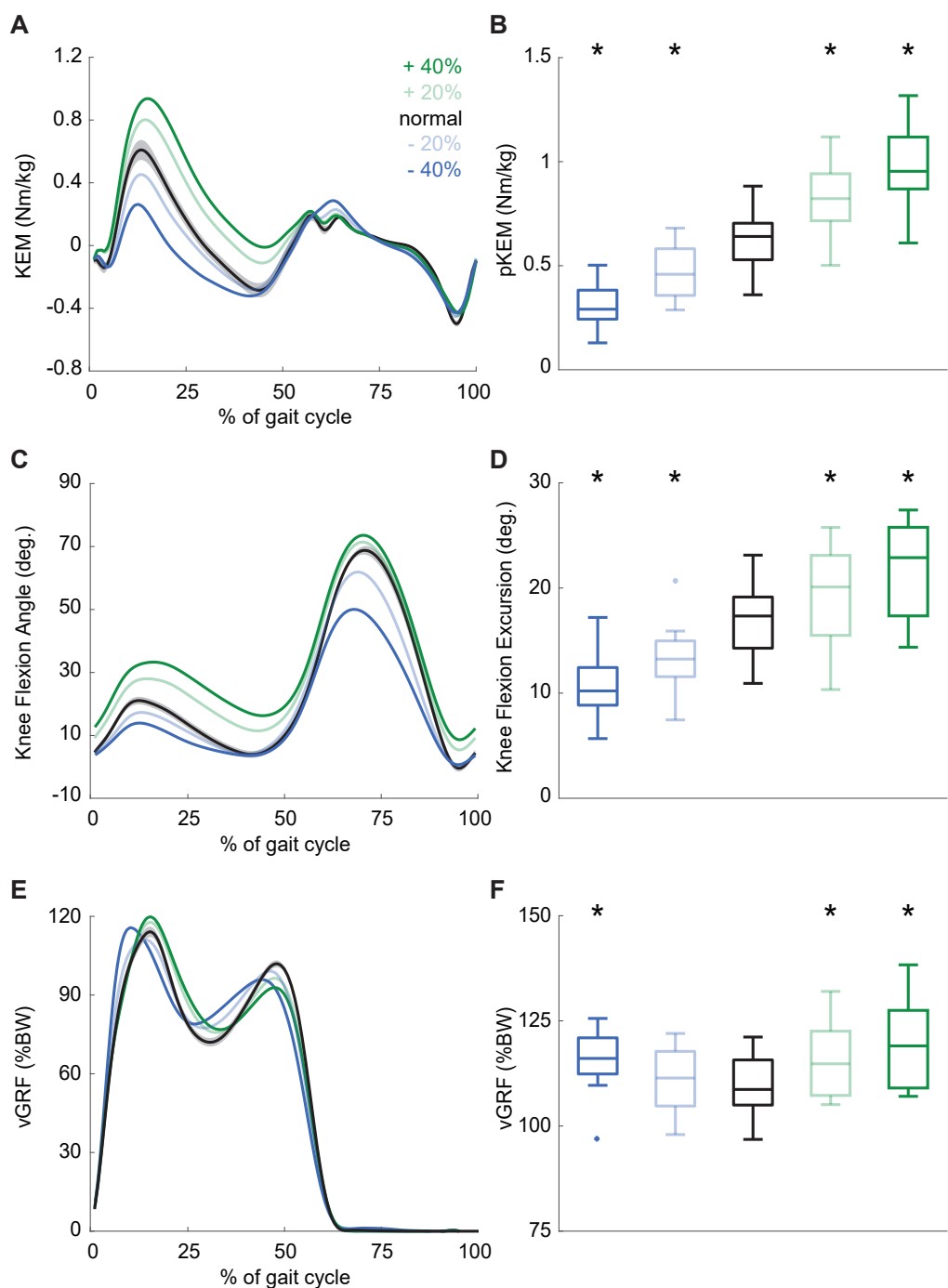

**Figure 2 Gait biomechanics as a function of time.** (A) Group mean knee extension moment plotted against an averaged gait cycle, from heel-strike to heel-strike. Gray shading represents the standard error for the normal walking condition. (B) peak knee extensor moment (pKEM) box plots across conditions. Asterisks (*) indicate a significant pairwise difference from normal walking. (C) Knee flexion angle normalized to the gait cycle. (D) Knee flexion excursion (instant of heel-strike to pKEM). (E) Vertical ground reaction force (vGRF) normalized to the gait cycle. (F) vGRF at instant of pKEM.

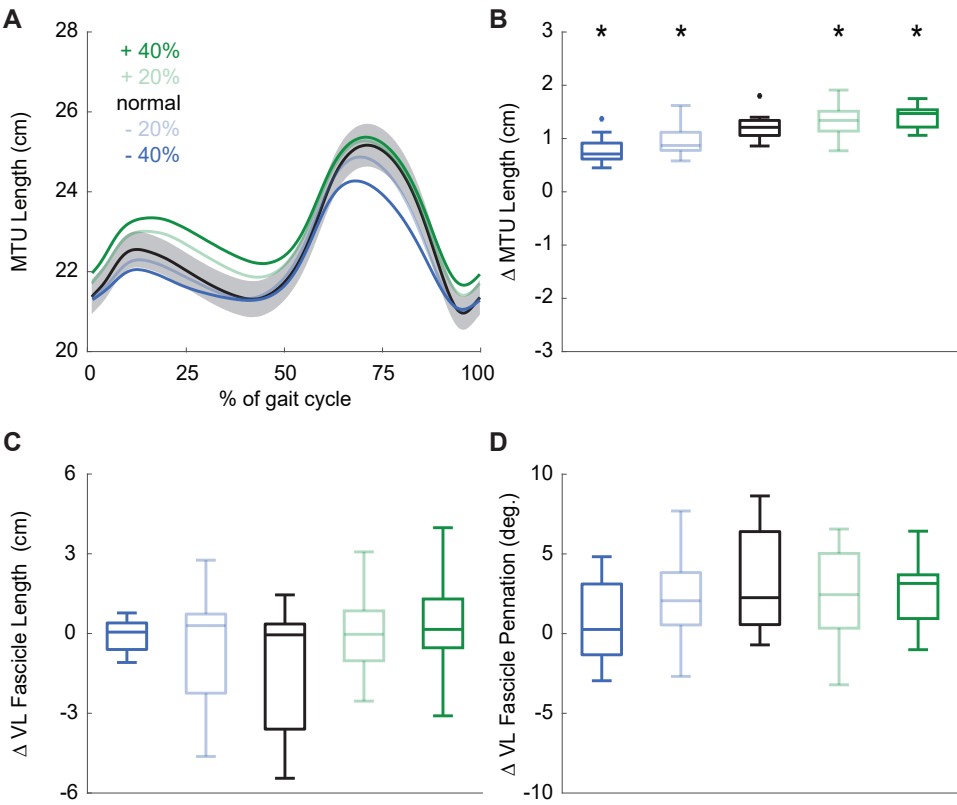

**Figure 3  Vastus lateralis muscle dynamics.** (A) Group mean vastus lateralis (VL) muscle-tendon unit (MTU) length plotted against an averaged gait cycle, from heel-strike to heel-strike. Gray shading represents the standard error for the normal walking condition. (B) Box plots for MTU length change between instants of heel-strike and peak knee extensor moment (pKEM) across conditions. (C) Box plots for VL fascicle length change between instants of heel-strike and pKEM across conditions. (D) Box plots for VL fascicle pennation change between instants of heel-strike and pKEM across conditions. Asterisks (*) indicate a significant pairwise difference from normal walking.

the potential for pKEM biofeedback to promote meaningful changes in gait biomechanics in the future application to individuals with ACLR and (2) provide benchmark *in vivo* data to better establish mechanistic links between quadriceps muscle dysfunction and altered knee joint biomechanics considered relevant to OA.

Knee extensor moments during walking, and changes thereof due to knee joint pathology, are routinely measured and reported in observational studies. These studies have demonstrated that, across a broad array of knee joint injuries and/or ligament reconstruction, quadriceps dysfunction and smaller pKEM during walking are prevalent compared to uninjured controls, even years after surgery and rehabilitation (*Mizner & Snyder-Mackler, 2005*; *Roewer, Di Stasi & Snyder-Mackler, 2011*; *Noehren et al., 2013*). Changes in gait biomechanics at the knee joint can shift articular contact forces to regions not conditioned to loading, particularly when the event allows little time for adaptation (*Andriacchi et al., 2004*). Our results demonstrate the capability to manipulate pKEM during walking, which may ultimately provide opportunities for intervention.

**Table 1** Vastus lateralis fascicle length outcome measures (mean ± SD).

| Condition | at HS (cm) | at pKEM (cm) | Δ length (cm) |
| --- | --- | --- | --- |
| −40% | 8.53 ± 3.04 | 8.46 ± 2.78 | −0.07 ± 0.64 |
| −20% | 8.53 ± 3.16 | 8.07 ± 2.85 | −0.52 ± 2.12 |
| Normal | 9.85 ± 3.49 | 8.54 ± 3.21 | −1.30 ± 2.32 |
| +20% | 9.31 ± 4.30 | 9.34 ± 3.46 | 0.02 ± 1.62 |
| +40% | 9.13 ± 3.96 | 9.55 ± 4.37 | 0.42 ± 1.85 |

**Notes.**

HS, Instant of heel-strike; pKEM, Instant of peak knee extensor moment.

In fact, the strategies participants used to modify their pKEM above and below their normal walking values were simple enough that a single ∼1-minute familiarization trial was sufficient to produce the observed changes during biofeedback trials. Clinical translation of pKEM biofeedback will rely on methodological advancements, as our approach leveraged sophisticated and expensive laboratory-based measurement equipment. However, advancements in wearable sensory technology (e.g., inertial measurement units *Hafer et al., 2020*) could provide a more practical means to prescribe pKEM biofeedback over multiple sessions in the clinic. After comparing our real-time estimates to inverse dynamics calculations of pKEM, we conclude that the higher than prescribed pKEM values demonstrated during biofeedback trials (i.e., +55% when cued with +40%) arose from small differences between our real-time surrogate model and inverse dynamic calculations, not from poor participant compliance. For example, our surrogate model neglects limb inertial effects. Indeed, the strong correlation and near linear association between real-time and inverse dynamics pKEM estimates supports the efficacy of our approach.

Based on the high prevalence with which reduced pKEM is accompanied by less knee flexion excursion in people with knee joint pathology, it is promising that the participants in this study consistently adjusted their pKEM via changes in knee flexion excursion during early stance. This kinematic change would subsequently alter the effective moment arm between the knee joint center and the GRF line of action. We also note that changes in knee flexion excursion in response to biofeedback were larger than the more modest changes in knee flexion angle at heel-strike, which increased only when targeting larger than normal pKEM (e.g., ∼8° for +40%). This suggests that participants maintained relatively normal flexion at heel-strike with adjustments thereafter during weight acceptance. Measured changes in peak vGRF are also unlikely to explain prescribed changes in pKEM across biofeedback conditions. Accordingly, we conclude that changes in knee flexion excursion are most responsible for changes in pKEM, especially when targeting smaller than normal values. Thus, this study provides evidence that pKEM biofeedback can promote desirable changes in both pKEM and KFE.

Real-time biofeedback applied in people with various knee joint pathologies have almost exclusively focused on augmenting peak vGRF (*Zeni Jr et al., 2013*; *Christiansen et al., 2015*; *Luc-Harkey et al., 2018a*; *Luc-Harkey et al., 2018b*). Both vGRF and pKEM biofeedback encourage individual participants to systematically manipulate their gait patterns, for example to optimize joint loading relevant to OA development. Indeed,

changes in limb loading are regularly accompanied by changes in the concentration of biomarkers relevant to cartilage health. For example, Luc-Harkey et al. showed that lesser peak vGRF in individuals with ACLR during walking associated with larger changes in serum concentrations of cartilage oligomeric matrix protein, a trend associated with cartilage thinning (*Erhart-Hledik et al., 2012*; *Luc-Harkey et al., 2018a*; *Luc-Harkey et al., 2018b*). It remains unclear how best to manipulate and thereby optimize knee joint loading during walking in individuals at risk of OA. However, as a more direct and thereby potentially improved surrogate for knee joint loading, additional studies that continue to leverage pKEM biofeedback are warranted. As an important next step, pKEM biofeedback should be tested in patient populations whose physical and psychological attributes may impact their ability to volitionally manipulate pKEM as described in this study.

As another major outcome of this study, our results contradict the textbook assumption that quadriceps MTU lengthening during gait is accompanied by eccentric muscle action. Not surprisingly, we found that the VL MTU lengthens considerably during weight acceptance. This MTU action coincides with the timing of knee flexion and significant quadriceps activation. We presume that these hallmark joint kinematic profiles and muscle activation explain the textbook assumption that the quadriceps muscles accommodate limb loading during early stance through eccentric action. However, our *in vivo* imaging results do not support this assumption. Indeed, we found that active VL muscle fascicles accommodate weight acceptance through relatively isometric action. To our knowledge, only two other studies have used ultrasonography to decouple fascicle and MTU dynamics during walking (*Chleboun et al., 2007*; *Bohm et al., 2018*). First, Chleboun and colleagues found that VL fascicles lengthened only 0.27 cm between 0% and 15% of the gait cycle despite 12.2° of knee flexion excursion (*Chleboun et al., 2007*). More recently, Bohm and colleagues used similar techniques and found 0.87 cm fascicle length change despite 1.81 cm MTU length change (*Bohm et al., 2018*). Consequently, we intuit that VL MTU lengthening during weight acceptance arises more from tendon elongation than from active muscle lengthening. Perhaps, as has been historically well-documented for MTUs spanning the ankle, isometric action of the quadriceps may be a fundamental phenomenon which may leverage elastic energy storage and return or to prevent muscle strain injury. Additional study in this area is warranted, especially given contemporary interest in isometric versus eccentric loading for tendon therapy (*Rio et al., 2015*).

Growing evidence of isometric action of VL muscles during human locomotion presents the additional opportunity to inform validation techniques for musculoskeletal simulations, especially given their use predicting knee joint loads (*Gardinier et al., 2014*; *Saxby et al., 2016*; *Wellsandt et al., 2016*). Isometric action of the plantarflexor muscles during walking (*Farris & Sawicki, 2012a*; *Farris & Sawicki, 2012b*) continues to encourage a reexamination of model parameters to better reconcile measurements with model predictions (*Arnold et al., 2013*). For example, when models incorrectly assume low tendon compliance, joint kinematics overshadow muscle activation and force-length-velocity relations to dictate estimates of muscle kinematics (*Arnold & Delp, 2011*). It is necessary that we decouple VL muscle–tendon dynamics to better estimate quadriceps force production and thus better

understand how changes in quadriceps function in those with knee joint injury affect the risk of OA development.

    This study has several limitations. First, we had to conduct normal walking trials before biofeedback trials in order to calculate target values. We also measured only right leg VL fascicle kinematics. Further, to promote reliability in our outcomes, we elected to measure fascicle lengths using manual tracking instead of automated tracking techniques (*Cronin et al., 2011*; *Farris & Lichtwark, 2016*). This decision has two potential limitations. First, we are unable to report on the time series of length change behavior that may occur during early stance. Second, we cannot conclusively state that the same fascicle was identified from all trials for each participant. It is also unclear if fascicle dynamics are consistent along the length of the VL, which could influence how well our muscle-level outcomes generalize. Finally, by design, our study focusses on sagittal plane knee joint kinematics, mechanics, and quadriceps muscle action; as well as the risk of cartilage degeneration due to loading below physiological values. However, individuals with knee joint pathology and those at risk of OA also frequently exhibit larger peak external knee adduction moments than controls (*Butler et al., 2009*;*Alnahdi, Zeni & Snyder-Mackler, 2011*), an indirect surrogate for medial compressive forces (*Ogaya et al., 2014*). Together, the collective literature thus suggests that *changes* in articular cartilage loading magnitude that occur faster than cartilage adaptation may contribute to PTOA (*Andriacchi et al., 2004*)—underscoring future opportunities for real-time biofeedback to optimize knee joint loading.

## CONCLUSIONS

In closing, we demonstrate that uninjured young adults can modulate pKEM during walking with concomitant changes in knee flexion excursion that are accommodated via relatively isometric, or even slight concentric, VL muscle action. Real-time pKEM biofeedback may be a useful rehabilitative and/or scientific tool to elicit desirable changes in knee joint biomechanics considered relevant to optimizing gait mechanics following knee injury.

### Funding

This work was supported by a grant from the NIH (R21AR074094) to Jason R. Franz and Brian Pietrosimone. The funders had no role in study design, data collection and analysis, decision to publish, or preparation of the manuscript.

### Grant Disclosures

The following grant information was disclosed by the authors:
NIH: R21AR074094.

### Competing Interests

The authors declare there are no competing interests.

## Author Contributions

- Amanda E. Munsch conceived and designed the experiments, performed the experiments, analyzed the data, prepared figures and/or tables, authored or reviewed drafts of the paper, and approved the final draft.
- Brian Pietrosimone conceived and designed the experiments, authored or reviewed drafts of the paper, and approved the final draft.
- Jason R. Franz conceived and designed the experiments, analyzed the data, prepared figures and/or tables, authored or reviewed drafts of the paper, and approved the final draft.

## Human Ethics

The following information was supplied relating to ethical approvals (i.e., approving body and any reference numbers):

Participants provided written informed consent and the UNC Biomedical Sciences Institutional Review Board approved this research (18-2185).

## Data Availability

The individual subject data underlying all measurements are available in the Supplementary File.

## Supplemental Information

Supplemental information for this article can be found online at http://dx.doi.org/10.7717/peerj.9509#supplemental-information.

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
