# Peer review of "The effects of knee extensor moment biofeedback on gait biomechanics and quadriceps contractile behavior"

_PeerJ, doi:10.7717/peerj.9509_

## Round 0.1 · original submission · Minor Revisions

Thank for your submission. This paper has now been reviewed by two experts in the field. As you will see, they have both suggested minor changes to the manuscript. Please consider these in your revision of the paper.

·

Basic reporting

The peak internal knee extensor moment (pKEM) is often lower in individuals with knee pathology than in healthy individuals and is a surrogate for quadriceps activity and cartilage loading. Because cartilage health is partially a product of loading environment, intervening to return pKEM towards normal magnitudes may be of value in the rehabilitation of knee pathologies. The purpose of the current study was to determine if and how young adults could alter pKEM during gait using real-time biofeedback. The authors developed a method for estimating pKEM in real-time and used this method to provide biofeedback to participants to target higher and lower than normal pKEM values. The authors examined whether these changes in pKEM resulted in changes in sagittal knee range of motion, vertical ground reaction force, and quadriceps MTU and fascicle kinematics. They found that changes in pKEM corresponded to changes in knee flexion excursion and MTU length change but not to fascicle length or angle change. They concluded that the reported methods and results lay the foundation for new intervention methods and for re-examining the presumed kinematics of the quadriceps muscles during loading response.

This paper is clearly written with established rationale, sufficient background information, and clear methods. Hypotheses follow logically from introduction, and methods and results correspond to hypotheses. Most of my comments are minor, with a few suggestions to improve clarity/reproducibility.

Raw data: Raw data to replicate the reported statistics are provided.

Experimental design

Experimental design corresponds to the aim of the study and appears technically sound.

1. KEM estimation method: Has agreement or correlation of this estimation method been compared to standard inverse dynamics KEM methods (i.e., bottom-up calculations)? Upon reading the results, I see that you did this comparison. Perhaps mention in the methods that this comparison was made or move this sentence of results (lines 209-212) to the methods.

2. The “experimental protocol” section is slightly unclear. Was the method for pKEM estimation used for both baseline target determination and for real-time biofeedback? During the exploration trial, were participants walking or standing still? See also comment 14 below on apparent mismatch between text and figure 1A.

3. How were key frame events identified in the ultrasound data? Frame closest in time to the events in the kinematic/kinetic data?

4. A figure would be helpful to accompany the description of the ultrasound methods. How was the linear projection of the aponeurosis defined? Were both deep and superficial aponeuroses always visible in both frames of interest? How was the deep aponeurosis applied as a correction factor? I think I can deduce the answers to these questions based on figures in some of the cited work, but a figure would greatly clarify this description.

5. Statistics: I am not familiar with determining if a change is different than no change by comparing it directly to 0 (VL fascicle length change isometric t-test). Was this recommended by a statistician?

Validity of the findings

Discussion elaborates on results in the context of previous work or assumptions. Future directions for this work are discussed. Limitations are appropriately outlined.

Additional comments

Overall, the manuscript is very well-written, has sound rationale, and is clearly executed. As such, my general comments are relatively minor.

6. Title: the title indicates that biofeedback affects quad contractile behavior but the results seem to indicate that the contractile behavior (i.e., change in fascicle length and angle) was similar across conditions. Suggest modifying title.

7. Abstract: VL is not defined before being used as an abbreviation.

8. Line 42: Suggest changing “arthroplasty” to “osteoarthritis”. There is no (or at least less) cartilage to load appropriately after arthroplasty.

9. First paragraph: Use of “physiological” loading term. I am not sure of the meaning being conveyed by this term. If in a functioning being, isn’t any magnitude of loading “physiological”? Is this meant to convey “normal” or “healthy”?

10. Lines 82-83: Is there a reference for “muscle activation does not associate with MTU behavior”?

11. Lines 100-101: Is there a measure of “quadriceps mechanical output” in this study? Or is the purpose to test a real-time biofeedback paradigm that very likely will affect quadriceps output? Suggest clarification of this, more precise wording, or a reminder that the pKEM being manipulated here very likely correlates with quad output.

12. Line 105: There is a hypothesis about muscle activation here but no matching methods or results. Remove?

13. Line 110: ( missing before “6 females”.

14. Lines 149-151: Figure 1A depicts the position vector as between the lateral femoral condyle and the instantaneous COP, which seems different than the description here of “between the lateral femoral condyle and the line of action of the GRF”. The latter makes more sense as the cross product of the GRF and the moment arm perpendicular to the GRF vector. Please confirm what method was used and ensure figure (and caption) match.

15. Line 176: “extension” or “extensor” moments?

16. Lines 176-177: Is the method described here the same cross product method as used for the biofeedback trials? Or this is traditional inverse dynamics?

17. Lines 219-223: The increase (decrease) reporting style took me a bit to figure out. Perhaps split this out to something like “For example, when cued to change pKEM by 40%, participants increased or decreased knee flexion excursion … by 30% and 36%, respectively”. Or use a similar style as in the following paragraph.

18. Lines 230, 234: standard deviations for fascicle shortening and pennation angle change?

19. MTU, fascicle results: how sensitive are these results to individual anthropometrics (e.g., height) or walking speed?

20. Lines 239-240: Similar to comment on purpose, what is the measure of “quadriceps output” in this study? Reframing similarly to purpose statement may add clarity to the application of the current findings.

21. Lines 241-242: Similar to pKEM methods questions, were the knee extensor moments reported in the results (as opposed to those used for biofeedback) not calculated using standard inverse dynamics methods? Were they expected to differ from typical profiles? Is this statement meant to convey “the people still walked like people when doing the biofeedback” or something else?

22. Line 265: Change “pEKM” to “pKEM”.

23. Lines 343-344: Does “cumulative literature” here refer to the literature on the cumulative effect of loading or the literature as a whole?

24. Table 1: Results are reported in mm here but in cm in the results section. Consider revising to match.

·

Basic reporting

Basic reporting is well done. I have checked all documents and have no concerns.

Experimental design

The experimental design is clear, appropriate, and in-line with the aims and scope of the journal. The methodology is especially detailed and the study could easily be replicated by experts in the field.

Validity of the findings

Again, this meets the journal's criteria. The study conclusions are valid and appropriate given the author's methodological approach and the results.

Additional comments

The purpose of this study was to examine the acute effects of biofeedback on gait biomechanics and quadriceps contractile behavior. The authors have concluded that biofeedback indeed influences knee extensor moment and knee flexion excursion, with isometric, rather than eccentric, muscle actions controlling these changes during weight acceptance. Both of these findings are very intriguing and perhaps not intuitive or widely known. In its current form, I believe that the manuscript is well-written and clear, with only a few issues needing clarification. The study seems well designed and the methodology appears detailed enough that the study could be replicated. The statistical analyses seem appropriate and the results are detailed. Some of the investigation’s potential limitations include a small sample size and analysis of a healthy, uninjured population despite the research topic being framed around knee pathology; however, the authors concede that this study was an initial step that may lead to larger, more definitive trials. Overall, I enjoyed reading this manuscript and I believe that this study has the potential to further the field’s understanding of the influence of biofeedback on gait parameters. My comments have been provided in chronological order.

Abstract
-While the sentence is well-written, the authors might consider better explaining what “±20% and ±40% of normal walking values” refers to. I suggest this because the authors examined a variety of different variables and readers may find this a bit unclear.
-Please spell out “vastus lateralis” before using VL.

Introduction
-Can the authors provide a reference to strengthen this statement? Lines 82-83: “However, muscle activation does not associate with underlying MTU behavior…”
-Line 102: “muscle-tendon unit” can be abbreviated as MTU both here and throughout the manuscript.

Materials and Methods
-The authors have been inconsistent with their use of “participants” versus “subjects.” Please pick one and be consistent.
-Line 110: Please include “(“ prior to 6.
-For readers that may not be familiar with the use of biofeedback during treadmill walking, a figure illustrating the authors’ laboratory setup would be quite helpful for gaining an appreciate of the participant’s perspective. Currently this is difficult to imagine from a written description alone.
-How confident are the authors that these participants were able to correctly adopt the recommended walking strategy in the absence of a separate familiarization trial? Are these altered gait mechanics simple enough to be adopted quickly?
-Lines 185-187: “We opted to perform manual identification of fascicle lengths and pennation at specific keyframe events (i.e., heel strike and the instant of pKEM) after determining that this approach was more reliable than automated tracking.” Can the authors provide any additional detail regarding this decision? Has anything on this been reported in the literature previously?
-It sounds like the authors approach to calculating fascicle length for those that fell outside the image window was similar to that recommended by Ando et al. (2014; JEK - https://www.ncbi.nlm.nih.gov/pubmed/24485560).
-While the box and whisker plots are appreciated, given the small sample size, I am curious if non-parametric tests would have been more appropriate for the dependent variables which were not normally distributed. Have the authors considered the Friedman test or other alternatives?
-Lines 198-202: The authors state that analyses were performed on five variables, but then list more than five. Please clarify.

Results
-The authors’ use of increase and decrease in lines 218-220 is difficult to follow.

Discussion section
-Very well-written and clear. I am just curious if the authors could elaborate on exactly how real-time biofeedback would be implemented in clinical practice, as well as how dosing would look? Is it likely that a single session or even repeated sessions would result in chronic changes in these gait parameters? I recognize that this is a preliminary study, but I guess I am missing the potential practical application associated with the intervention.

Clear and helpful figures; the table is appropriate.

---

## Round 0.2 · accepted · Accept

Thank you very much for your considered revision to the original submission. I appreciate the time you took to update the paper as required and for providing a detailed response to the reviewer comments. I look forward to seeing this paper in print. Congratulations!